# A Rad51-independent pathway promotes single-strand template repair in gene editing

Danielle N. Gallagher[1], Nhung Pham[2], Annie M. Tsai[1], Nicolas V. Janto[1], Jihyun Choi[1], Grzegorz Ira[2], James E. Haber[1]*

1 Department of Biology and Rosenstiel Basic Medical Sciences Research Center, Brandeis University, Waltham, MA, United States of America, 2 Department of Molecular and Human Genetics, Baylor College of Medicine, Houston, TX, United States of America

* haber@brandeis.edu

**Data Availability Statement:** All relevant data are within the manuscript and its Supporting Information files.

## Abstract

The Rad51/RecA family of recombinases perform a critical function in typical repair of double-strand breaks (DSBs): strand invasion of a resected DSB end into a homologous double-stranded DNA (dsDNA) template sequence to initiate repair. However, repair of a DSB using single stranded DNA (ssDNA) as a template, a common method of CRISPR/Cas9-mediated gene editing, is Rad51-independent. We have analyzed the genetic requirements for these Rad51-independent events in *Saccharomyces cerevisiae* by creating a DSB with the site-specific HO endonuclease and repairing the DSB with 80-nt single-stranded oligonucleotides (ssODNs), and confirmed these results by Cas9-mediated DSBs in combination with a bacterial retron system that produces ssDNA templates *in vivo*. We show that single strand template repair (SSTR), is dependent on Rad52, Rad59, Srs2 and the Mre11-Rad50-Xrs2 (MRX) complex, but unlike other Rad51-independent recombination events, independent of Rdh54. We show that Rad59 acts to alleviate the inhibition of Rad51 on Rad52's strand annealing activity both in SSTR and in single strand annealing (SSA). Gene editing is Rad51-dependent when double-stranded oligonucleotides of the same size and sequence are introduced as templates. The assimilation of mismatches during gene editing is dependent on the activity of Msh2, which acts very differently on the 3' side of the ssODN which can anneal directly to the resected DSB end compared to the 5' end. In addition DNA polymerase Polδ's 3' to 5' proofreading activity frequently excises a mismatch very close to the 3' end of the template. We further report that SSTR is accompanied by as much as a 600-fold increase in mutations in regions adjacent to the sequences directly undergoing repair. These DNA polymerase ζ-dependent mutations may compromise the accuracy of gene editing.

## Author summary

DNA double strand breaks (DSBs) are one of the most lethal types of damage that can be inflicted on a chromosome and failure to repair such lesions can result in chromosome instability, commonly associated with human cancer. A knowledge of DNA repair

**Funding:** This work has been supported by the National Institute of Health grants R35 GM127029 to J.E.H and GM080600 and GM125650 to G.I. D. N.G. has been supported by NIGMS Genetics Training Grant T32GM007122 (https://www.nigms.nih.gov/) and by the National Science Foundation Graduate Research Fellowship Program under grant 1744555 (https://www.nsfgrfp.org/). A.J. was supported by a Research Education for Undergraduates (REU) grant from the National Science Foundation (https://www.nsf.gov/crssprgm/reu/). The funders had no role in study design, data collection and analysis, decision to publish, or preparation of the manuscript.

**Competing interests:** The authors have declared that no competing interests exist.

mechanisms is also critical in the exploitation of gene therapy, a process that includes intentionally breaking the DNA to modify the genetic sequence. Here we compared two site-specific methods to create DSBs (HO endonuclease and CRISPR/Cas9) in budding yeast, to modify several DNA targets by single-strand DNA template repair (SSTR). We show that gene editing uses a DSB repair pathway that is independent of the canonical repair protein Rad51 and distinct from previously studied Rad51-independent pathways in its requirements for several other known recombination proteins. We show both in gene editing and in single-strand annealing that Rad59 acts to suppress the modulation of Rad52's strand annealing activity by Rad51. We also determined how mismatches in the template are incorporated into the genome, and that this assimilation reflects different aspects of Msh2-mediated mismatch repair as well as Polδ-mediated proofreading. These insights provide insight into the mechanisms of DSB repair by this important gene-editing pathway.

## Introduction

DSBs are repaired through one of two pathways: homologous recombination (HR) or nonhomologous end joining (NHEJ). Both classical NHEJ and microhomology-mediated end joining (MMEJ) involve DNA ligase-mediated joining of the broken chromosome ends, which usually results in small insertions or deletions (indels) at the junction [1,2,3,4,5,6]. HR is a less mutagenic form of DSB repair, as it makes use of a homologous sequence as a donor template for repair. The template can be located on a sister chromatid, a homologous chromosome, or at an ectopic site. The majority of HR events are dependent on a core group of proteins, including the Rad51 strand-exchange protein that is responsible for homology recognition and initiating strand invasion into a double-stranded DNA (dsDNA) template [7]. In budding yeast, Rad51 interacts with and is assisted by several key recombination proteins, including the mediator Rad52 and the Rad51 paralogs, Rad55 and Rad57, as well as the chromatin remodeler, Rad54 [8,9,10,11,12,13]. Rad52 also plays a critical role in later steps of DSB repair, facilitating second-end capture of the DNA polymerase-extended repair intermediate [14].

However, some DSB repair events, though still requiring Rad52, are Rad51-independent. The best studied mechanism is single-strand annealing (SSA), where homologous sequences flanking a DSB are rendered single-stranded by 5' to 3' exonucleases and then annealed, creating genomic deletions [15,16,17,18]. SSA requires the Rad52 paralog Rad59, especially when the size of the flanking homologous regions is small [19]. A second Rad51-independent mechanism involves break-induced replication (BIR) [20,21]. Rad51-independent BIR is also independent of Rad54, Rad55, and Rad57; however, Rad59 and a paralog of Rad54 called Rdh54/Tid1 assume important roles. Rad51-independent BIR also requires the Mre11-Rad50-Xrs2 complex, whereas Rad51-mediated events and SSA are merely delayed by the absence of these proteins [18]. A third Rad51-independent pathway, another form of BIR, operates to maintain telomeres in the absence of telomerase (known as Type II events). Here too, Rad59 and the MRX complex, as well as Rad52, are necessary, whereas the Type I Rad51-dependent telomere maintenance pathway does not require either Rad59 or MRX [22,23,24]. Both Rad51-dependent and Rad51-independent forms of telomere maintenance require the nonessential DNA polymerase δ subunit, Pol32, as do other BIR events [25] Similarly, DSB repair by intramolecular gene conversion involving short (33-bp) regions of homology is inhibited by Rad51 and is dependent on the MRX complex, Rad59, and Rdh54 [26]. Rad51-independent BIR pathways

are also dependent on the Srs2 helicase that antagonizes loading of Rad51 onto resected DSB ends [21,27].

Use of the RNA-guided CRISPR/Cas9 endonuclease has revolutionized gene editing in eukaryotic systems ranging from yeast to mammals [28,29,30]. Guided endonucleases are programmed to create site-specific DSBs that can be repaired by providing a homologous template [31,32]. One approach that has been shown to be an efficient method of gene editing in a variety of eukaryotic systems is to introduce short single-stranded oligodeoxynucleotides (ssODN) as a donor template [33,34,35,36].

Here we have examined the genetic requirements for single strand template repair (SSTR) in budding yeast, using two different systems: 1) an inducible HO endonuclease and an 80-nucleotide (nt) ssODN as a template for DSB repair, and 2) an optimized bacterial retron system to produce ssDNA templates *in vivo* with a targeted Cas9-mediated DSB. We confirm that in budding yeast, as in other eukaryotes, SSTR is a Rad51-independent mechanism, but show that this pathway is distinct from the previously-described Rad51-independent recombination pathways. SSTR depends on Rad52, Rad59, and Srs2 proteins, as well as the MRX complex, but is independent of Rdh54/Tid1. Surprisingly, deleting Rad51 suppresses the *rad59Δ* defect. We show a similar suppression of *rad59Δ* by *rad51Δ* in SSA. We conclude that Rad59 prevents Rad51 from inhibiting Rad52-mediated strand annealing. In contrast, this novel form of repair is specific to ssDNA, as dsDNA templates of the same size and sequence use a canonical Rad51-dependent process.

By analyzing the fate of mismatches between the ssODN and the target DNA, we show that the mismatches at the 5' and 3' ends of the template are differently incorporated into the gene-edited product. We show that both the *MSH2* mismatch repair (MMR) protein and the 3' to 5' exonuclease activity of DNA Polδ play important roles in resolving heteroduplex DNA. Finally, we demonstrate that SSTR is accompanied by as much as a 600-fold increase in mutations in the 1-kb region adjacent to the site of gene editing. These mutations are dependent on the error-prone DNA polymerase ζ that fills in single-stranded regions generated during DSB repair.

## Results

### Single stranded template repair is Rad51-independent

Gene editing using ssODNs in yeast is limited both by the efficiency of DSB initiation and by the efficiency of transformation to introduce the ssDNA template. As a model for DSB-induced gene editing, we used a galactose-inducible HO endonuclease to create a site-specific DNA break at the *MATα* locus of chromosome 3, coupled with the introduction of the ssDNA template by transformation. In this strain, both *HML* and *HMR* donors have been deleted, so that a DSB that can only be repaired via NHEJ unless an ectopic donor is provided [1]. When HO is continually expressed, imprecise NHEJ repair occurs in approximately $2 \times 10^{-3}$ cells, distinguishable by indels in the cleavage site that prevent further HO activity. Repair by homologous recombination can be accomplished by introducing an 80-nt ssODN as a repair template. Cells were transformed with an ssODN template and then plated onto media containing galactose, which rapidly induces an HO-mediated DSB [37]. The template contains 37-nt of perfect homology to each end of the DSB, surrounding a 6-nt *Xho*I restriction site (Fig 1A). SSTR leads to the disruption of the HO cleavage site by the insertion of the *Xho*I site, whose presence can be confirmed by an *Xho*I digest of a PCR product spanning the region (S1 Fig). In WT cells, we achieve an editing efficiency of 75–90% among in DSB survivors, with the remaining survivors repaired via NHEJ; these indels are eliminated by mutants such as *mre11Δ*, *rad50Δ*, and *yku70Δ* that are known to be required for NHEJ (Fig 1B). However,

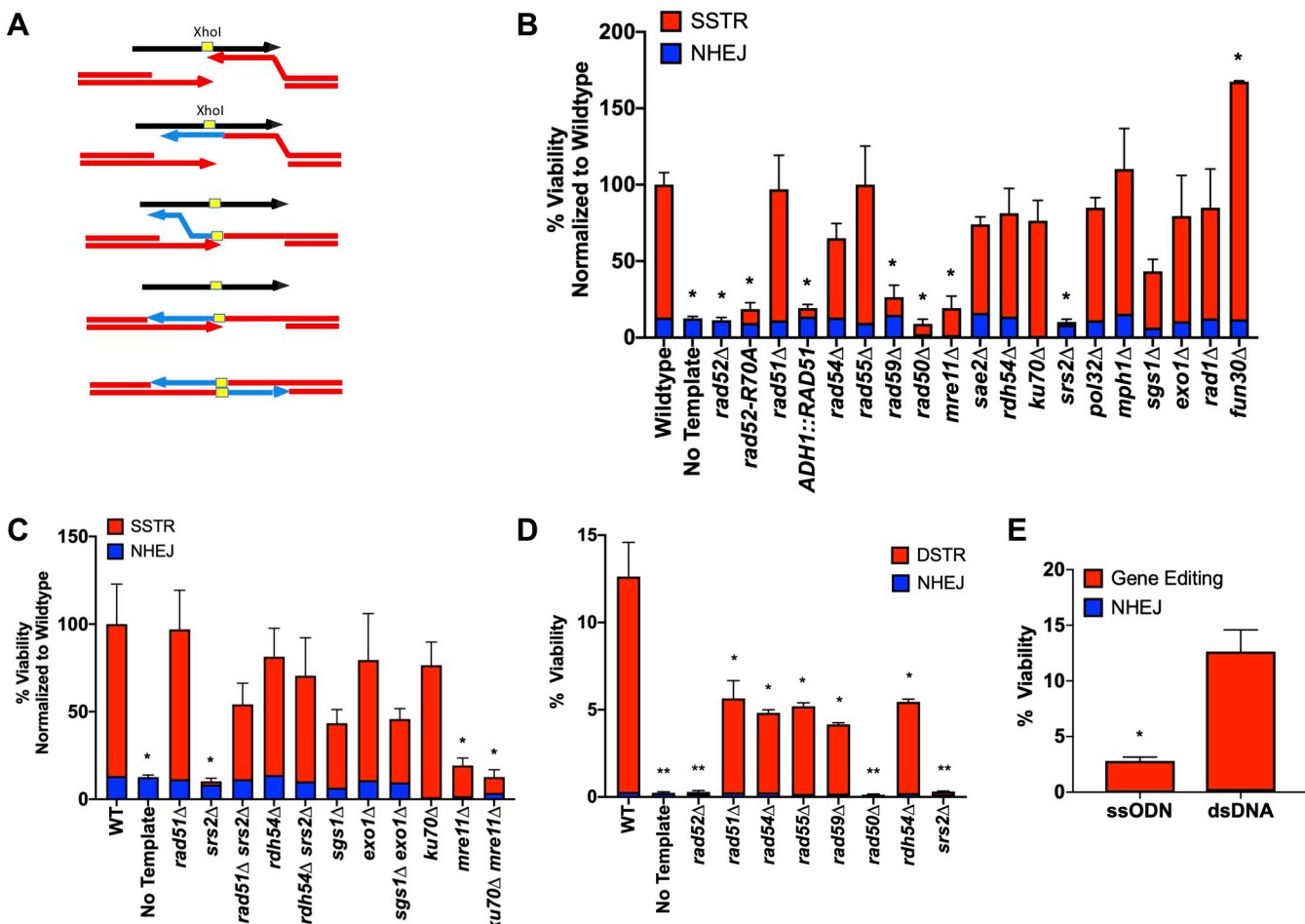

**Fig 1. SSTR is a novel form of Rad51-independent DSB repair.** A) A DSB was created at the *MATα* locus via induction of a galactose-inducible HO endonuclease. A ssODN with 37-nt homology on either side of an *Xho*I site (yellow) provides a template for SSTR, resulting in the insertion of an *Xho*I restriction site into the *MAT* locus. B) Viability determined by plate counts from galactose-containing media (induction media) over YEPD (non-induction media). Proportion of SSTR (red) and NHEJ (blue) events in various mutants involved in DSB repair are determined via PCR of the *MAT* locus, followed by an *Xho*I digest (S1 Fig). C) Effect of double mutants on SSTR. D) Genetic requirements to repair an HO-mediated DSB with an 80-bp dsDNA repair template. D-E) Viability tests to determine strains ability to undergo DSB repair with designate 80-nt ssDNA or 80-bp dsDNA template. Viability was determined as described in B F) Viability comparison of ssODN to dsDNA. Significance determined using a t-test with Welch's correction, * p ≤ 0.01, ** p ≤ 0.001, comparing mutant's to WT. Error bars refer to standard error of the mean. WT and *rad51Δ* n = 9, all other mutants n = 3 for SSTR, n = 3 for all dsDNA assays.

survival in this assay is quite low, with an average survival rate of 2.8% (S2 Fig); this low efficiency reflects limitations in transforming ssODN, as shown below.

We applied this assay to determine which recombination factors are required for SSTR. Consistent with previous results in both budding yeast and metazoans [35,36,38,39], SSTR proved to be Rad51-independent (Fig 1B). Furthermore, SSTR was significantly inhibited when Rad51 was overexpressed from an *ADH1* promoter on a multicopy 2*μ* plasmid. SSTR proved to be independent of the Rad51 paralog, Rad55, and the Rad54 translocase/chromatin remodeler, both of which are required for most DSB repair events that involve dsDNA templates. Also, the Shu complex, which contains Rad51 paralogs and has been shown to promote error-free HR, was also dispensable for SSTR [40]. As expected, SSTR was dependent on the single strand annealing protein Rad52, with essentially all *rad52Δ* survivors resulting from NHEJ events. There was also a significant reduction of SSTR in the absence of the Rad52

paralog Rad59, as well as less profound reduction in a strain lacking the helicase Sgs1, although *sgs1Δ* did not prove to be statistically significant in this assay (Fig 1B). However, since this assay compares a large number of mutants, we used a strict statistical p-value of 0.01 and *sgs1Δ* has a p-value of 0.014 when looking at the total cell viability. If we compare only SSTR events, *sgs1Δ* has a p-value of 0.02 compared to WT, still not meeting our cut-off for statistical significance.

In previous studies of Rad51-independent recombination, excluding SSA which is also Rad51-independent, both Rdh54 and the MRX complex were required [21,26]. For SSTR, while MRX is required, Rdh54 is not, distinguishing SSTR from previously studied Rad51-independent pathways. SSTR is also independent of Pol32. It is also notable that while the MRX complex is required, Sae2 –which often functions in conjunction with MRX in regulating end-resection but not in DSB end-tethering or other functions [41, 42, 43]–is not. A *sae2Δ* strain is still capable of both SSTR and NHEJ.

SSTR also requires the Srs2 helicase (Fig 1B). One major function of the Srs2 helicase is to act as an anti-recombination factor by stripping Rad51 from the ssDNA tails formed after resection [44,45]; thus, *srs2Δ* might mimic the inhibition of SSTR that is seen when Rad51 is overexpressed. Indeed, deleting *RAD51* suppressed *srs2Δ*'s defect in SSTR (Fig 1C). Moreover, deleting *RDH54* suppressed the defect in *srs2Δ*. Although these results might suggest that Rdh54 acts in the same pathway as Rad51, we do not believe this to be the case since their deletions behave differently in other genetic combinations (see below).

We also examined the role of several genes that are involved in the 5' to 3' resection of DSB ends: the Exo1 exonuclease and the Sgs1-Rmi1-Top3-Dna2 helicase/endonuclease complex [46,47]. Deleting either Sgs1 or Exo1 had no significant effect on the efficiency of *Xho*I insertion, and neither did the *sgs1Δ exo1Δ* double mutant (Fig 1C). These results suggest that the MRX complex can provide sufficient end resection to allow SSTR involving the homologous 37 nt of the ssODN donor. Deleting the Fun30 SWI/SNF chromatin remodeler has also been shown to strongly retard 5' to 3' resection of DSB ends [48,49,50], but we found a significant increase in SSTR. We note that in previous research using human cancer lines, SSTR was dependent on proteins in the Fanconi anemia (FA) pathway [51]. The helicase function of Mph1 is the only homolog of the FA pathway found in yeast, but Mph1 does not appear to play a role in SSTR in our system (Fig 1B).

Since our SSTR assays employ templates containing only 37-nt of homology on either side of the DSB, we wanted to know if this noncanonical repair pathway is specific to ssDNA, or might also apply to repair with dsDNA templates with the same limited homology. We annealed complementary 80-nt ssDNA oligonucleotides to create a dsDNA template with free ends that had 37-bp of perfect homology flanking each side of a 6-bp *Xho*I restriction site. After duplexing, the pool of dsDNA template was treated with S1 nuclease to degrade any remaining non-duplexed ssDNA. We transformed the template into cells using the same protocol used with the ssDNA templates (Fig 1D). With this short dsDNA template, the repair process shifted to a Rad51-dependent event, now also requiring Rad54 and Rad55, but still dependent on the MRX complex, Srs2 and Rad59 (Fig 1D). DSTR is also dependent on Rdh54, whereas most DSB Rad51-dependent repair events are not [52, 53, 54]. Double-stranded template repair (DSTR) proved to be more than five times as efficient as SSTR (Fig 1E). It is important to note that when the duplexed ssODNs were not S1-treated so that the template pool still contained un-annealed ssDNA, the process is Rad51-independent. A residual amount of ssDNA after S1 treatment may explain why we don't see a stronger dependence on Rad51 in this assay.

## Rad59 regulates Rad52-mediated strand annealing

Given that Rad59 has an important role in SSTR, we further examined the genetic interaction between Rad51 and Rad59. Previous biochemical studies had [55] had suggested that Rad59 might mediate the ability of the Rad52 protein to facilitate single-strand annealing, which is the first step in SSTR (Fig 1A), specifically by modulating the inhibitory effect of Rad51 on Rad52's strand annealing activity. Indeed, deleting Rad51 suppressed the inhibition of SSTR by *rad59Δ* and increased the rate of SSTR significantly higher than observed in WT cell or in the absence of Rad51 (Fig 2A). We then examined a separation-of-function mutation of Rad52, *rad52-R70A*, that is proficient for loading of Rad51 onto ssDNA but fails to carry out strand annealing [56]. SSTR is severely impaired in a *rad52-R70A* mutant, similar to *rad59Δ*'s phenotype (Fig 1B; Fig 2A). However, deletion of Rad51 did not suppress *rad52-R70A*, supporting the conclusion that Rad59 affects Rad51's modulation of Rad52-mediated strand annealing activity. Since deletion of *RAD51* in a *rad59Δ* or a *rdh54Δ* background had a marked increase on the cells ability to undergo SSTR, we also asked if *rad59Δ rdh54Δ* might show a similar increase in SSTR. However, *rad59Δ* is not suppressed by *rdh54Δ* (Fig 2A).

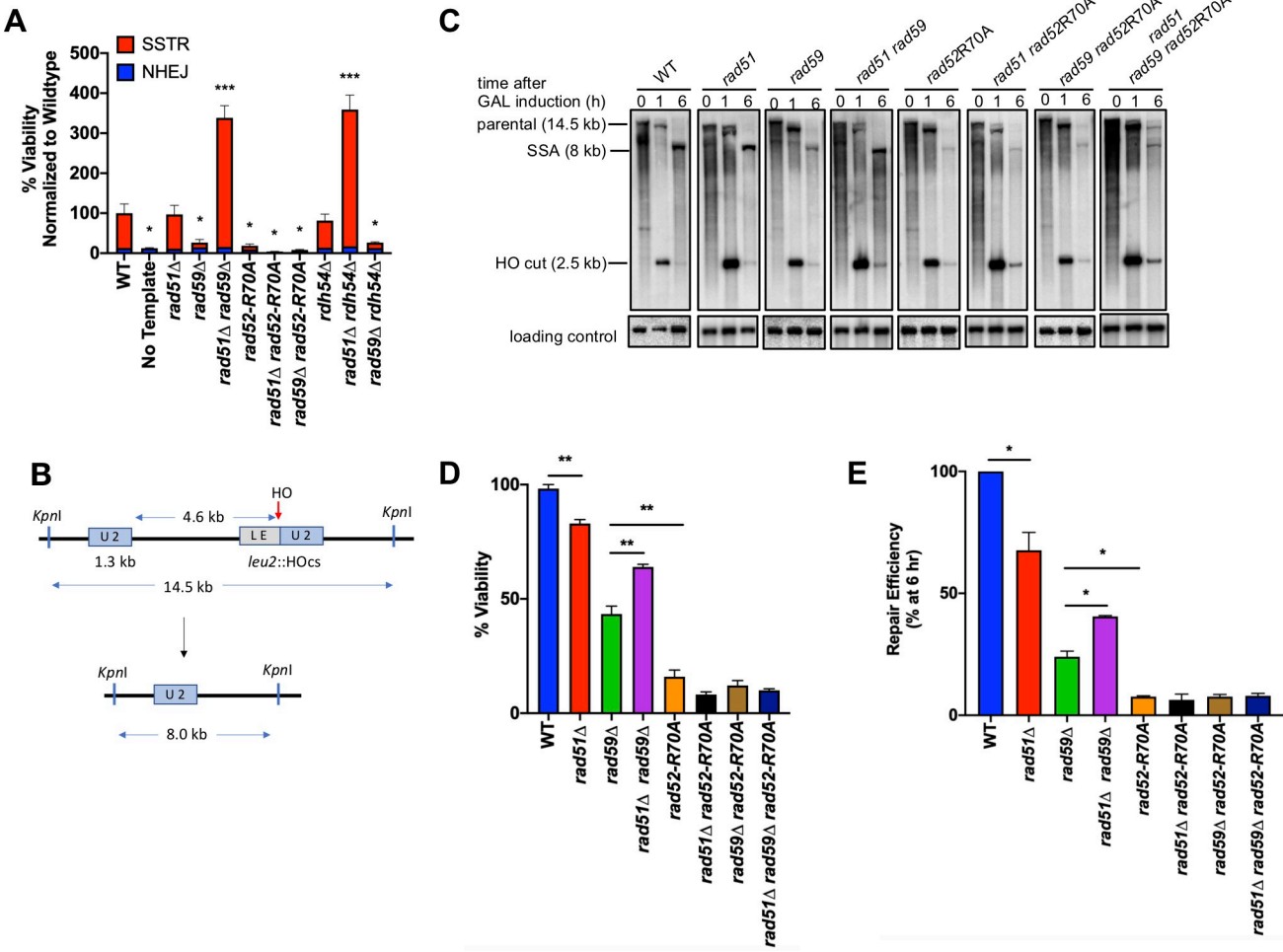

**Fig 2. Rad59 alleviates the inhibition of Rad51 on annealing activity of Rad52.** A) Viability of single and double mutants following SSTR, determined by plating on galactose-containing media. B) An HO-induced DSB results in repair via SSA between partial *leu2* gene repeats located 5 kb apart on the left arm of chromosome 3. C) Representative southern blots showing DSB repair products by SSA in WT and indicated mutants. D) Viability of mutants on galactose-containing plates, where HO DSBs are repaired via SSA (mean ± SD; n = 3). Welch's t-test was used to determine the p-value. E) Graphs show quantitative densitometric analysis of repair efficiency by 6 hr compared to WT (mean ± SD; n = 3). Welch's t-test was used to determine the p-value.

To confirm that Rad59 affects the strand-annealing function of Rad52, we turned to a well-characterized SSA system, in which a DSB promotes formation of a deletion between flanking repeated sequences [57]. Previous studies have shown that Rad59 is important in SSA, especially when the length of the flanking homologous repeats is short, below a few hundred base-pairs [19]. We used an HO-induced DSB within a *leu2* gene, which is repaired by SSA with a direct "*U2*" repeat (1.3 kb) located 5 kb away (Fig 2B), resulting in a chromosomal deletion of the sequences located between the two repeats, as well as one of the partial copies of the *leu2* gene. Like SSTR, SSA is severely impaired in a *rad52-R70A* mutant (Fig 2C; S2 Fig). Deletion of Rad59 reduced the efficiency of SSA of the 1.3-kb repeats to approximately 30% (Fig 2D). Deletion of Rad51 by itself only has a mild negative effect on SSA, but deletion of Rad51 partially suppressed the SSA deficiency of *rad59Δ* (Fig 2D). As with SSTR, this suppression was not observed in either *rad52-R70A rad51Δ* or *rad52-R70A rad59Δ*, again suggesting that Rad59 specifically plays a role in the modulation by Rad51 of Rad52-mediated strand anneal-ing; however we did not see the same large increase in the *rad51Δ rad59Δ* that we do in SSTR, suggesting that Rad59 might have an additional role in SSTR initiated by the HO-endonucle-ase or when there is limited homology in the donor sequence.

## Polζ is responsible for target-adjacent mutagenesis

Previous research has suggested that DSB repair that involves DNA resection is highly muta-genic because ssDNA regions created during resection must be filled in once HR has com-pleted [58,59,60]. Gap-filling, either by DNA polymerase δ or by translesion DNA polymerases such as Polζ have been shown to be responsible for mutation rates 1000-fold over background spontaneous mutation rates in canonical DSB repair [61,62]. Since SSTR is a form of HR and likely involves extensive end-resection and gap-filling, it seemed possible that there are significant off-target effects that have not previously been considered. To examine this pos-sibility, the yeast *URA3* gene within an MX cassette was inserted 200 bp centromere-proximal to the HO cleavage site, such that the *URA3* sequences themselves are approximately 400 bp beyond the 37 nt of homology shared between the DSB end and the ssODN template. These cells were then targeted in the same *Xho*I insertion assay previously described (Fig 3A). SSTR survivors were collected and then replica-plated onto 5-fluroorotic acid media (FOA), which selects for *ura3* mutants [63]. Compared to the spontaneous mutation rate of *URA3* mutations ($3.5 \times 10^{-7}$), determined by fluctuation analysis [64], there was an almost 600-fold increase in *ura3* mutants after gene editing events (Fig 3B). However, this increased mutagenesis is con-fined to a region close to the area of the DSB, as the rate of mutagenesis drops significantly as the *URA3* marker was inserted further upstream. For a site that is approximately 650 bp upstream of the DSB, there was an approximate 50-fold increase in mutagenesis, whereas at 2.2 kb there was only an 8-fold increase. There was a nearly 200-fold increase in *ura3* muta-tions when the *URA3* marker is located 550 bp downstream of the DSB. We could not investi-gate the effects of moving the *ura3* marker further downstream of the DSB given that Taf2, an essential gene, is located 3' of the HO cleavage site. There is a statistically significant difference between integrating *URA3* upstream and downstream of the HO cleavage site. It is possible that this difference is simply due to the increased distance, however, it raises the possibility the two sides of the DSB engage different repair machinery, discussed below.

The source of the frequent *ura3* mutations appears to be dependent on the error-prone DNA polymerase Polζ, as deleting either the Rev1 or Rev3 components of Polζ, resulted in very few *ura3* mutants (Fig 3C). However, neither deletion of Rev1 nor Rev3 affects cell viability, indicating that other, less mutagenic mechanisms can be used for gap-filling in the absence of Polζ (S3 Fig). When using dsDNA of the same size and sequence, we found an

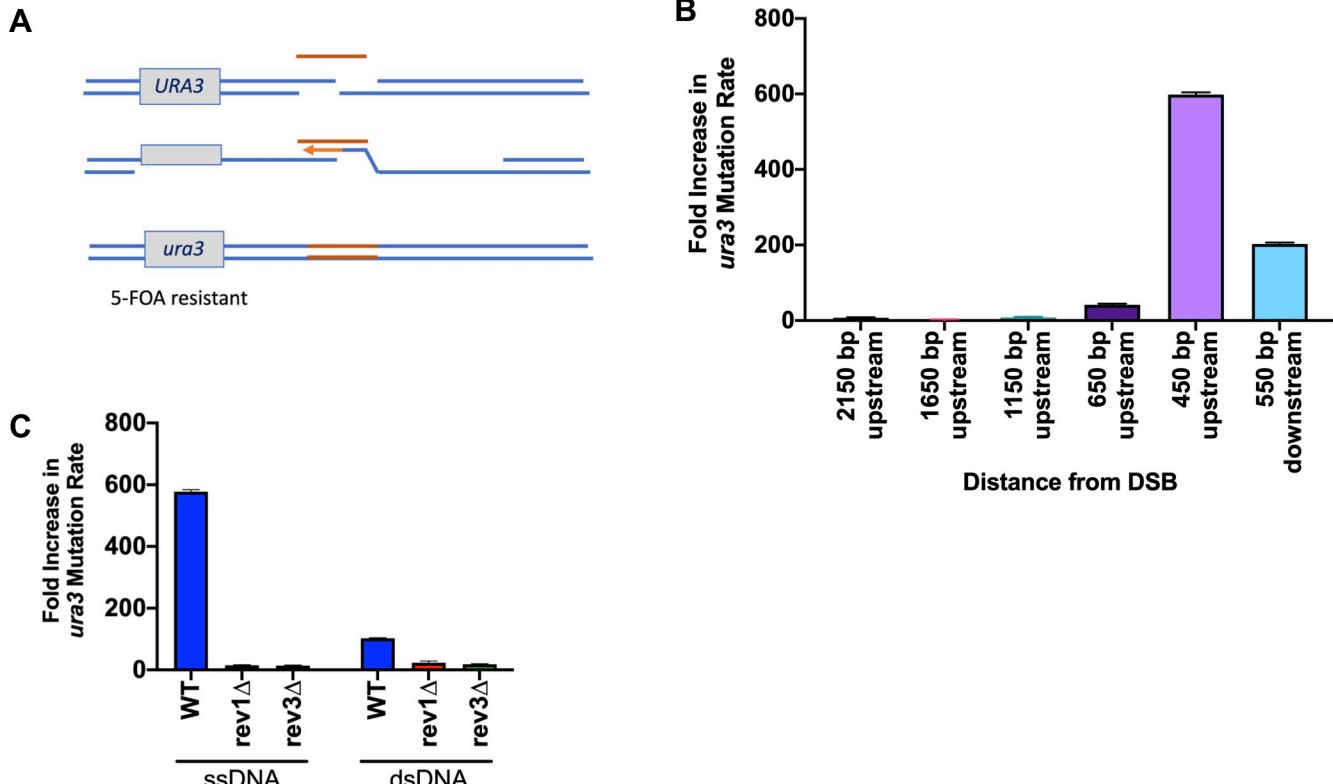

**Fig 3. *Target-adjacent mutagenesis is dependent on* Polζ.** A) Effect of SSTR on a *URA3* gene integrated near the site of HO cleavage. Mutations in *URA3* following SSTR were collected by replica plating survivors on galactose media onto 5-FOA medium. B) The increase in mutation rate in SSTR over the spontaneous mutation rate (determined by a fluctuation analysis) was determined at the indicated locations surrounding the DSB site. C) Effect of deleting Rev1 and Rev3 components of Polζ. Significance determined using an unpaired t-test with Welch's correction, comparing mutants to WT. Error bars refer to standard error of the mean. Spontaneous mutation rate determine by fluctuation analysis, n = 10. SSTR mutation rate, n = 3.

approximately 100-fold increase in target-adjacent mutagenesis (Fig 3C). These mutagenic events are still dependent on the activities Polζ. These data suggest that adjacent off-target effects of gene editing pose a danger that should be ruled out in selecting gene-editing events.

## Additional genetic requirements of SSTR depend on template design

To test if changing the design of the ssODN donor altered the genetic requirements of SSTR, we used a ssODN similar to that described in Fig 1A, except that the 37 nt of homology on each side of the *Xho*I site were each targeted to sequences that are 500 bp from the DSB; thus, successful gene editing via the 80-nt ssODN creates a 1-kb deletion flanking the *Xho*I site (Fig 4A). Successful SSTR should then only occur after extensive 5' to 3' resection of the DSB ends. Repair efficiency, as measured by viability, using this donor template was significantly lower than the ssODN that simply incorporated an *Xho*I restriction site (approximately 1% compared to 3%) (S4 Fig).

The core recombination requirements of SSTR for this configuration were the same as those seen with the simple *Xho*I insertion, as gene editing was independent of Rad51, Rad54, Rad55, Rdh54, and Sae2, but still dependent on Rad52, Rad59, Srs2, and the MRX complex (Fig 4B). As before, *srs2Δ* was suppressed by both *rad51Δ* and *rdh54Δ*, and there were still substantial increases in *rad51Δ rad59Δ* and *rad51Δ rdh54Δ* compared to wildtype or *rad51Δ* (Fig 4B and 4C). Moreover, using an ssODN that creates a large deletion imposes additional

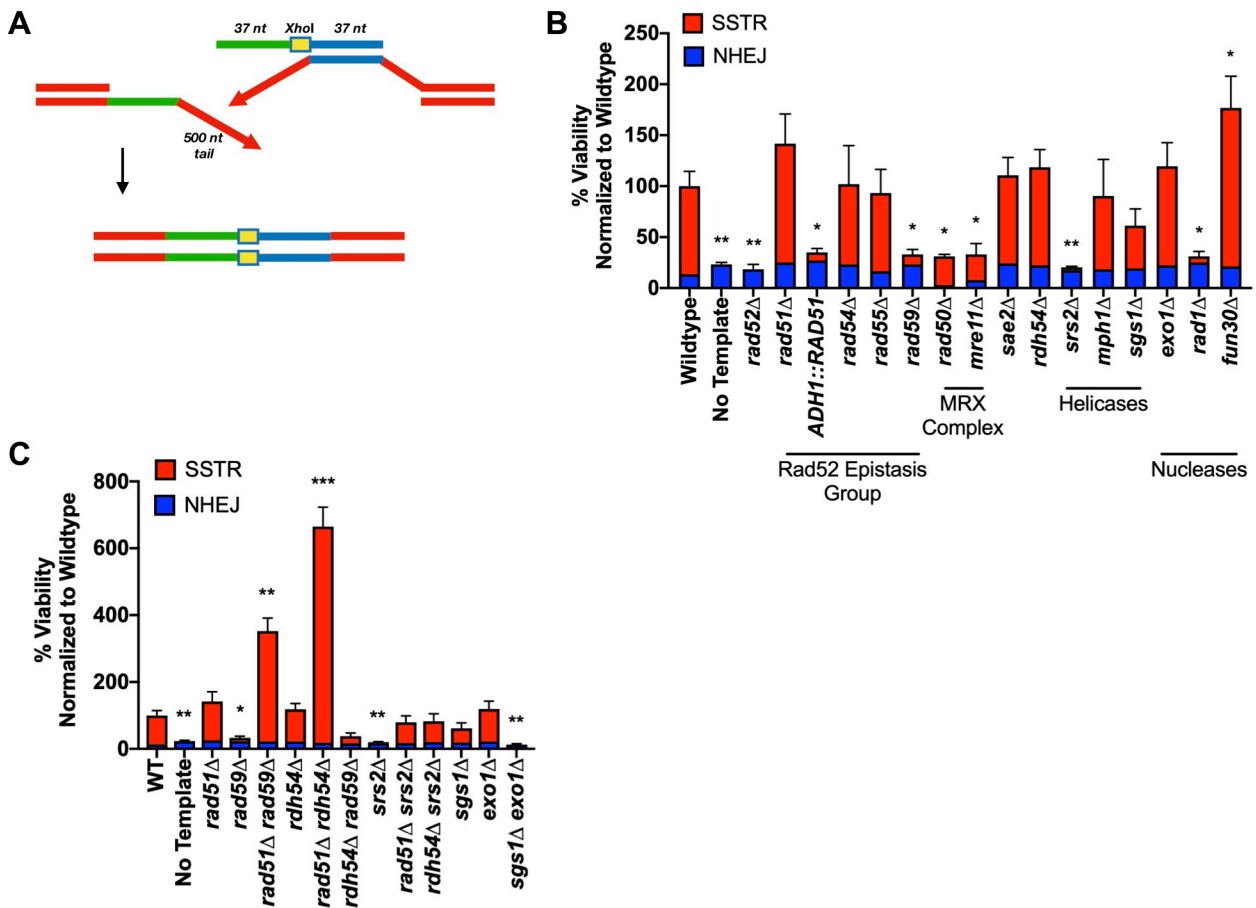

**Fig 4. The genetic requirements of SSTR are dependent upon ssODN template design.** A) SSTR using a ssODN with 37-nt homologies located 500 bp on either side of the HO-induced DSB. SSTR results in a 1-kb deletion and the incorporation of the *Xho*I restriction site into the *MAT* locus, which can be screened via PCR and restriction digest with *Xho*I. B) Viability determined by galactose-induction plate counts over YEPD plate counts. Proportion of SSTR and NHEJ events in various recombination mutants determined via PCR (S2 Table). C) Effect of double mutants on deletions created by SSTR. Significance determined using an unpaired t-test with Welch's correction comparing mutant's to WT, * $p \leq 0.01$, ** $p \leq 0.001$, *** $p \leq 0.0001$. Error bars refer to the standard error of the mean. WT and *rad51*Δ n = 9, all other mutants n = 3.

requirements. For the ssODN to pair with a resected DSB end, there must be extensive 5' to 3' resection. While deleting either the Sgs1 or Exo1 individually had no significant impact, the double mutant *sgs1*Δ *exo1*Δ abolished SSTR (Fig 4C). This result stands in contrast to the lack of effect of the double mutant in the simple incorporation of the *Xho*I site and emphasizes the need for long-range 5' to 3' resection (Fig 1C). The effect of blocking long-range resection in *sgs1*Δ *exo1*Δ was not mimicked by deleting Fun30, whereas in previous studies examining 5' to 3' resection of DSB ends *fun30*Δ significantly slowed resection similar to *sgs1*Δ *exo1*Δ [48, 49, 50]. Previously, deleting Fun30 was shown to protect double-stranded DNA fragments used in "ends-out" transformation [48], but in the present scenario, the transformed DNA is single-stranded. Whether Fun30 also affects the stability of ssODN's is not clear.

Another requirement in the deletion assay is for Rad1, and presumably Rad10, which together act as a 3' flap endonuclease that can remove the 3'-ended 500-nt nonhomologous tail that would be created by annealing the ssODN to its complementary strand [16, 65] (Fig 4A). Rad1 is not needed in the simple *Xho*I insertion assay (Fig 1B).

## Genetic requirements of SSTR are identical using Cas9

Although it was convenient to survey many mutations using the highly efficient and easily inducible HO endonuclease, we confirmed that the same genetic requirements apply when a DSB is created by CRISPR/Cas9. To overcome the low efficiency of transforming ssODNs into yeast and to better screen Cas9-mediated gene editing events, we turned to a modified version of the CRISPEY system to produce ssDNA templates *in vivo* [66]. This system utilizes a yeast-optimized *E. coli* retron system, Ec86, to generate designer ssDNA sequences *in vivo*. Retrons are natural DNA elements encoding for a reverse transcriptase (RT) that acts on a specific consensus sequence to generate single stranded DNA products [67,68,69]. These ssDNA products are covalently tethered to their template RNA by the RT, however after reverse transcription the RNA template is degraded [70,71]. The CRISPEY system utilizes a chimeric RNA of Ec86 joined to the gRNA scaffolding of Cas9 at the 3' end [66]. By integrating a yeast-optimized galactose-inducible Cas9 and retron (RT) onto chromosome 15, and using a CEN/ARS plasmid containing a gRNA linked to the retron donor template (Figs 5A and S6), we were able to achieve high efficiency of Cas9-mediated gene editing at two different chromosomal locations, within the *MAT* locus near the HO cleavage site, and at a 5-bp insertion in the *lys5* locus (Fig 5A and 5B). Compared to the <3% of cells that properly inserted the *Xho*I site at *MAT* with an HO-induced DSB and a transformed ssODN template, the retron system yielded efficiencies of >20%. At *lys5*, successful SSTR via an 80-nt retron-generated ssDNA donor carrying the wild type *LYS5* sequence results in Lys$^+$ recombinants that are easily recovered at a rate of 34%, compared to the <1% of lysine prototrophic events in cells with the same Cas9-induced DSB, but in the absence of the retron donor to provide an ssDNA repair template (Fig 5B and 5C; S5 Fig).

To test whether the genetic components of SSTR were the same with a Cas9-induced DSB, we introduced gene deletions in this strain background. The requirements were generally the same as for HO-induced SSTR events, being independent of Rad51, Rad55, and Fun30, but still dependent on Rad52, Rad59, Srs2, and the MRX complex (Figs 5C and S5). With the retron system and Cas9 endonuclease, the double mutants *rad51Δ rad59Δ* and *rad51Δ rdh54Δ* do not show the same significant increase in SSTR above WT levels as we observed with the HO-endonuclease. This difference could be explained by several different reasons. First, the 3' end of the retron ssDNA is covalently linked to the gRNA, so the template itself may limit access to the repair machinery from the 3' end of the template. In addition, the tethering of the template to Cas9 could change the dynamics of the homology search [72]. There could be other differences as well, as Cas9 may stay bound to DNA after cleavage, although how long it remains bound *in vivo* is not clear [73, 74].

## Mismatch repair acts differently at the 5' and 3' ends of the ssODN

How SSTR occurs is still not fully understood. One question concerns the fate of the ssDNA template strand itself. Another concerns the fate of mismatches between the template strand and the complementary single-stranded DSB end. We used the same ssODN described in Fig 1, but now using ssODNs that contained mismatches in the donor sequence. One donor had 4 mismatches 5' to the *Xho*I site, while the second had 4 heterologies on the 3' side, spaced every 9-nt (Fig 6A). We note that mismatch position 1 is only 2-nt away from the end of the ssODN. We observed that there was a significant decrease in gene editing with 4 mismatches on either the 5' or the 3' side, compared to the fully homologous template, although the effect was more pronounced on the 3' side (Fig 6B). This difference may reflect the fact that the initial annealing steps in SSTR can only happen on the side 3' to the *Xho*I site and may be quite different from the capture of the second end.

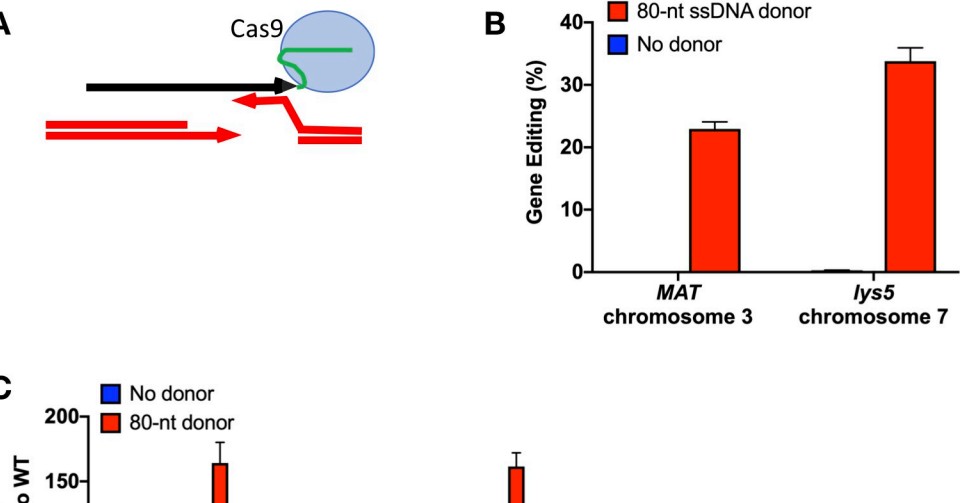

**Fig 5. SSTR initiated by retron-Cas9 utilizes a Rad51-indepndent repair pathway.** A) The retron system utilizes a modified Cas9 gRNA that tethers the ssDNA donor template to the RNA scaffolding of the Cas9 protein. Successful SSTR at the *MAT* locus results in the insertion of an *Xho*I restriction site, as in Fig 1. SSTR at the *lys5* locus repairs a 5-bp insertion in the *lys5* locus, resulting in Lys$^+$ recombinants. B) Efficiency of the Retron-Cas9 system at two chromosomal locations. Cells were plated onto URA$^-$ plates with dextrose (non-induction) and URA$^-$ with galactose (induction) media. At *MAT*, the percent gene editing was determined by PCR and *Xho*I digest of induction survivors as described in Methods. At *lys5*, the percentage of gene editing was determined by replica plating URA-Gal survivors onto Lys$^-$ media. The resulting plate count over plate counts of URA$^-$ non-induction media results in % gene editing. C) Effect of recombination mutants on retron-Cas9 SSTR gene editing. After induction of the retron system on galactose-containing media, survivors were replica plates to Lys$^-$ media. The frequency of Lys$^+$ colonies was calculated as a percentage of total cells plated and normalized to wild type. Significance was determined using two-tailed t-tests compared to WT, using the two-stage Benjamini, Krieger, and Yekutieli false discovery rate approach [89], $^*$ p $\leq$ 0.01, $^{**}$ p $\leq$ 0.001, $^{***}$ p $\leq$ 0.0001. Error bars refer to the standard error of the mean. n = 3. *rad59Δ* compared to *rad59Δ rad51Δ* p = 0.009.

In studies of SSA and DSB-mediated gene conversion, the inhibitory effects of a small percentage of mismatches could be overcome by deleting Sgs1 or components of the mismatch repair system [75,76]. Here, however, deleting Sgs1 did not suppress the effect of the four mismatches. In fact, with mismatches on the 3' side of the *Xho*I site, the majority of survivors in *sgs1Δ* were NHEJ events. Deleting Sgs1 also consistently reduced SSTR in the fully homologous case, though not statistically significantly in any one assay (Figs 1 and 4).

Deleting the mismatch repair gene, *MSH2*, did not suppress the reduced level of SSTR in the templates carrying 4 mismatches (Fig 6B), but there was a notable change in the inheritance of these mismatches (Fig 6C). We analyzed the DNA sequences of 23 SSTR events in both wild type and *msh2Δ* strains for each of the ssODNs carrying mismatches (Fig 6C). In wild type strains, a majority of *Xho*I insertions were accompanied by co-inheritance of 3 of the 4 mismatches. However, in an *msh2Δ* strain, the majority of SSTR events using the 5'

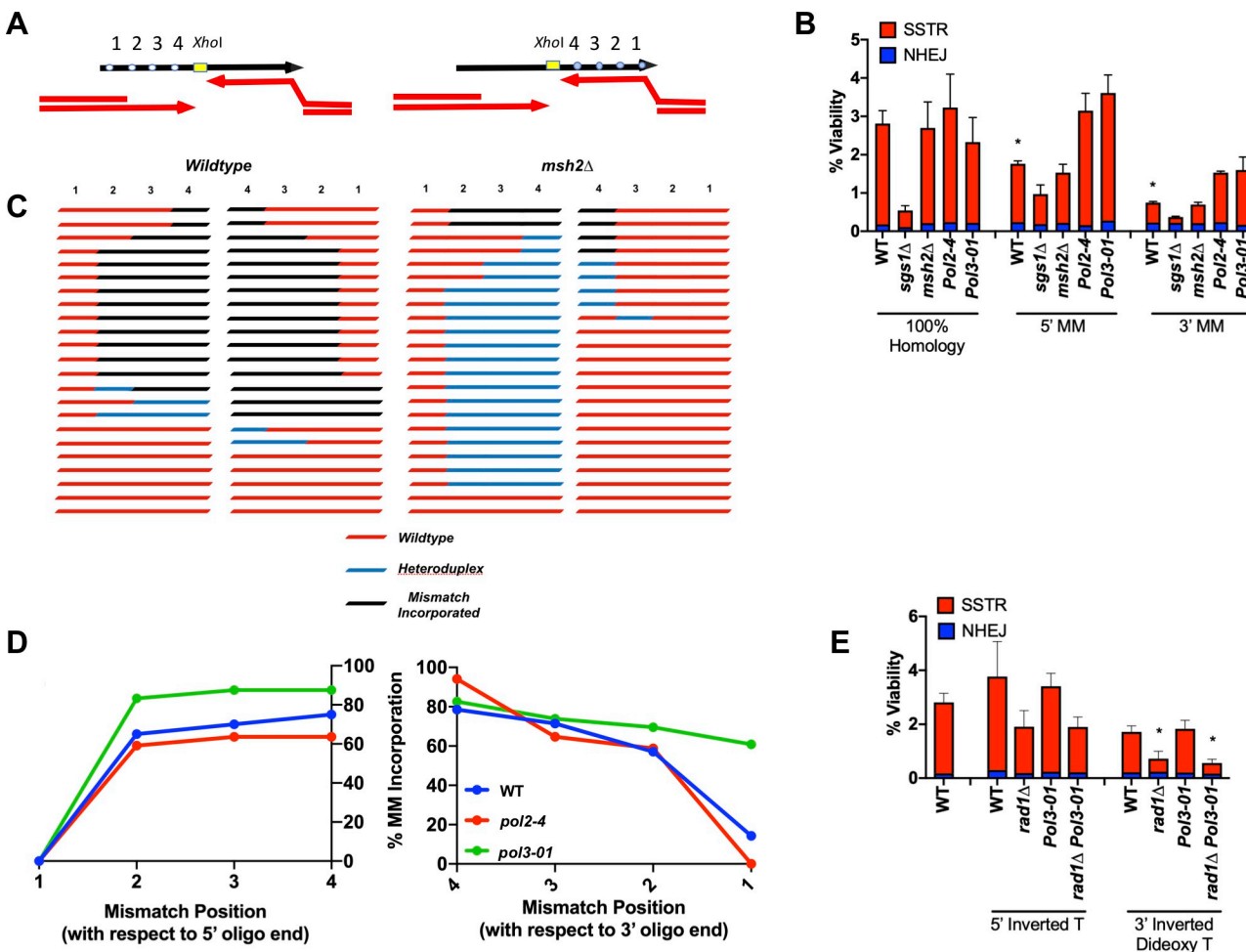

**Fig 6. Gene editing events are dependent on the activity of MSH2.** A) Experimental set-up to determine the effect of mismatches on SSTR. Mismatches are spaced every 9 nt from the *Xho*I site in the center of the template. Mismatch 1 is located only 2 nt from the terminus of the ssODN. SSTR events are determined by incorporation of an *Xho*I restriction site. B) Significance determined using an unpaired t-test compared to WT of the same template, * p ≤ 0.01. n = 3. C) Inheritance of mismatches in wild-type (left) and *msh2*Δ (right) strains. The inheritance of markers was determined separately by sequencing SSTR survivors using ssODNs with mismatches 5' or 3' of the *Xho*I site. The different outcomes were grouped together for comparison. D) Sequencing n = 23 per template per genetic background. E) Viability test of cells ability to survive DSB repair via SSTR with a template containing a chemical modification on the terminus of the ssODN. Significance determined using an unpaired t-test with wild-type levels using a template that does not have chemically modified ends, * p ≤ 0.01. Error bars refer to standard error of the mean. n = 3.

mismatch ssODN template showed heteroduplex tracts, as evidenced by the presence of both the chromosomal and mutant alleles at these sites when DNA from single transformant colonies were sequenced. On the 3' side, *msh2*Δ eliminated the great majority of events in which the mutations in the ssODN were inherited into the gene-edited product. These results extend the conclusions reached by Harmsen *et al.* studying SSTR in mammalian cells, where the absence of mismatch repair largely prevented incorporation of heterologies on the 3' half of the ssODN, while not preventing their assimilation in the 5' half [77]. In their studies of mammalian cells, it was not possible to detect the presence of unrepaired heteroduplex DNA, as we show in Fig 6.

We noted that mismatches located 2-nt from either end of the ssODN were only rarely assimilated into the gene-edited product (Fig 6C). In our recent study of break-induced replication (BIR), we discovered that heteroduplex DNA created by strand invasion was corrected (i.e. mutations were assimilated into the BIR product) in a strongly polar fashion from the 3' invading end [78]. Moreover, these corrections of the heteroduplex were orchestrated by the 3'

to 5' exonuclease (proofreading) activity of DNA polymerase δ, which removed up to 40 nt from the invading end and replaced them by copying the template. Incorporations of the mismatched base from the template was almost completely abrogated by eliminating the proofreading activity of DNA polymerase δ (*pol3-01*). Here, using the *Xho*I insertion ssODN with 4 mismatches on one side or the other, we found that the overall-incorporation of mismatches was unaffected by proofreading-defective mutations in Polε (*pol2-4*) or Polδ (*pol3-01*), with one notable exception: the mutation 2-nt from the 3' end of the template was incorporated at a very high level in *pol3-01* mutants (Fig 6D). These data suggest that Polδ can be loaded not only onto the 3' end of the chromosomal DSB, where it initiates copying of the rest of the ssODN template, but can also be recruited by the 3' DNA end of the ssODN itself and then chew back the 3' end. It is possible that Polδ might also extend this end of the ssODN template and raises the possibility that the ssODN itself could be incorporated into the gene-edited product in some cases. The *pol3-01* mutation did not affect the assimilation of the most terminal mismatch on the 5' end of the ssODN (Fig 6D).

### Effect of modifying the 5' and 3' ends of the ssODN

Work by Harmsen *et al.* in mammalian cells also showed that blocking the ends of the template strand reduced SSTR, suggesting that the ssDNA strand itself might be more than a simple template that anneals with a DSB end and is then copied by a DNA polymerase [77]. If the ssODN might be assimilated into the product, then blocking access to either 5' or 3' end of the template might affect its usage. We used the *Xho*I insertion ssODN with complete homology to the chromosomal site, except that these templates were chemically modified with either an inverted thymine at the 5' end, or an inverted dideoxy-thymine on the 3' end. These chemical modifications should prevent ligation of the ssODN into chromosomal DNA, since the terminal thymines can't pair with the resected chromosomal DNA, and should block extension of the 3' end by Polδ unless the block is excised. There was no significant difference between either modified and unmodified donor templates in wildtype cells or in a *pol3-01* strain. An alternative way that a modified 3' end nucleotide might be removed would be through the use of the Rad1-Rad10 flap endonuclease. Indeed, we found a modest reduction in SSTR with the 3' block in a *rad1Δ* strain when compared to wildtype cells (Fig 6E). These results indicate that the template might sometimes be ligated into the repaired product, or that the inverted thymine causes increased rejection of the ssODN as a suitable repair template.

We propose a model for SSTR where edits templated by the 5' and the 3' are incorporated through different mechanisms. Incorporation of mismatches on the 3' end of the *Xho*I site should occur only during the time that the resected DSB end has paired with the donor, to prime DNA polymerase to copy the template strand (Fig 7). On the 5' side, however, the initial copying of the template and its subsequent annealing to the second DSB end should obligately produce heteroduplex DNA that will be resolved by mismatch repair to be fully mutant or fully wild type, dependent on the activity of Msh2 (Fig 7). We suggest that the failure to incorporate edits located close to the 5' end into the final product might occur if the DNA polymerase copying the template dissociates before it has copied the last several nucleotides, so this site is not incorporated as heteroduplex DNA involving the second DSB end; alternatively, there could be a Msh2- and DNA polymerase proofreading-independent mechanism that corrects the heteroduplex in favor of the chromosomal sequence (Fig 7).

### Discussion

Although CRISPR/Cas9 has made it possible to generate specific changes to the genome in many organisms, budding yeast still serves as an important resource to determine the

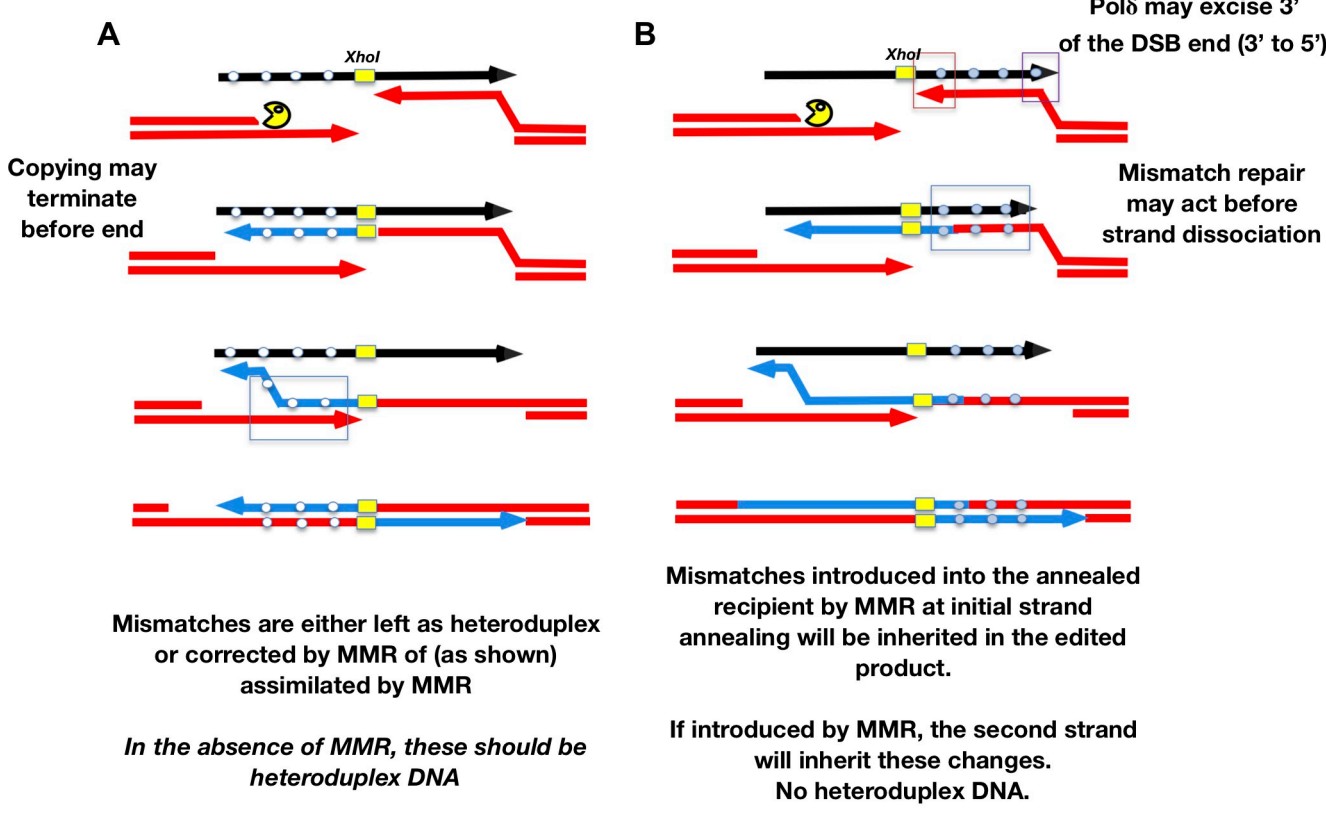

**Fig 7. Model of heteroduplex formation and incorporation of mismatches during SSTR.** Fate of an ssODN with 4 mismatches either 5' to the *Xho*I site (A) or 3' (B). After the DSB is created and resected only the strand on the right can anneal to the template. This annealing creates a heteroduplex that may be repaired by the mismatch repair machinery including Msh2. Heterologies close to the 3' end of the invading strand, but also at the 3' end of the ssODN, can be excised by the 3' to 5' exonuclease activity of DNA polymerase δ. Only if the heteroduplex is converted to the template strand genotype will these mismatches be incorporated into the SSTR product (B). Mismatches 5' to the *Xho*I site will be obligately copied by DNA polymerase after strand invasion (A). The dissociation of the newly copied strand allows it to anneal with the resected second end of the DSB, creating an obligate heteroduplex. Dissociation of the newly copied strand may occur without copying the end of the ssODN template. Heteroduplex DNA may then be corrected to the genotype of the donor template or left as unrepaired heteroduplex. In the absence of Msh2, most outcomes will have heteroduplex to the left of the *Xho*I site but no incorporation to the right, resulting in sectored colonies.

mechanism of gene editing, and thus to optimize experimental design. We show that gene editing using ssODN templates utilizes a novel pathway of DSB repair that is independent of the canonical DSB repair proteins Rad51, Rad54, Rad55 and Rdh54/Tid1, but still depends on Rad52, Rad59, Srs2, and the MRX/MRN complex. When the ssODN templates are designed to create a large deletion, Rad1-Rad10 flap endonuclease and the long-range resection machinery also become essential. We have confirmed that the genetic requirements of SSTR are generally the same in HO-mediated SSTR as with Cas9-mediated events. Moreover, the mechanism of SSTR is specific to ssDNA templates, as gene editing using a dsDNA template of the same size and sequence (with 37 bp homology to each DSB end) switches to a Rad51-dependent mechanism that is nevertheless distinct from that involving gene conversion between long regions of homology, most notably by its requirement for Rdh54 and for the MRX complex. Although Rad51-independent BIR events require Pol32, it also is not necessary in SSTR, possibly because of the length of new DNA synthesis can be accomplished without additional processivity factors. The role of Sgs1 remains to be investigated.

We have also determined an important role of the Rad52 paralog Rad59, which apparently acts to alleviate the inhibition of Rad51 on Rad52's ability to anneal ssDNA tails. Previous studies have shown that Rad51 impairs Rad52's ability to anneal DNA strands, but this can be overcome by Rad59 [54]. In both SSTR and SSA, deleting Rad51 suppresses *rad59Δ*. We note that suppression of *rad59Δ* by *rad51Δ* is quite a different relationship than is seen in spontaneous recombination between chromosomal regions, where *rad51Δ rad59Δ* is much more severe than *rad59Δ* or *rad51Δ* alone [78]. In some of our assays the efficiency of SSTR is significantly greater for *rad51Δ rad59Δ* than for *rad51Δ* alone; this result suggests that Rad59 may impair SSTR in other ways than simply modulating Rad51. Rad59 interacts directly with Rad52 and may form heteromeric rings [79, 80]; possibly without Rad59, Rad52's annealing activity is intrinsically greater. We note that Rad59 is also important in DSTR with very short homology, while its role in DSB repair involving chromosomal regions with much longer homology is not critical [21]. There are two distinct strand-annealing steps in SSTR: the initial annealing between the resected end of the DSB and the 3' end of the ssODN template, followed by second-end capture in which the newly copied strand anneals to the other resected end of the DSB. In DSTR, Rad51 is apparently needed for the initial invasion of the resected end into the dsDNA template, but there would still be a second-end capture process where Rad59 might be critical when the homology is very short (in our experiments, only 37 nt). Similarly, the Srs2 helicase is essential for SSTR (and DSTR). The role of Srs2 as an anti-recombinase to displace Rad51 appears to accounts for its importance, as *rad51Δ* suppresses *srs2Δ*.

We do not yet understand the role of *RDH54*. In other contexts, HO endonuclease-induced Rad51-independent recombination, involving either intrachromosomal plasmid recombination or interchromosomal BIR events, requires Rdh54 [26]; but in SSTR and DSTR, Rdh54 is not required. Recent studies have suggested that a major role for Rdh54 is in controlling the size of a strand invasion D-loop [52]. However, in SSTR there is no D-loop but only an annealed structure between the 3' end of the ssODN and the resected DSB end. How such strand invasions occur in Rad51-indpendent events remains unknown, but apparently Rdh54 is important in facilitating this event. Why *rdh54Δ* suppresses *srs2Δ* in SSTR also remains unclear. Without Srs2, more Rad51 will be loaded onto the ssDNA end of the DSB, and potentially also on the ssODN, and stabilization of the Rad51 filament structure when the regions are short may depend on Rdh54. Rdh54 also plays a novel role in interchromosomal template switching during DSB repair, where a partially copied strand of DNA jumps from one template to another [54,81]. These secondary jumps are impaired in *rdh54Δ* but simple gene conversion events, copying one ectopic template, are not. Possibly the dissociation of the newly copied first strand in SSTR from its short template (Fig 7) requires this template-jumping activity.

SSTR shares some features with other HR pathways that involve short regions of homology [22, 26], such as the need for the MRX complex. In DSB repair events that involve longer (>200 bp) homology, deleting components of MRX delays but does not diminish HO-induced DSB repair [82]; yet with short substrates MRX plays a central role, both for SSTR and DSTR. Yeast MRX has been implicated in many early steps in DSB repair [83]; it is required for most NHEJ events, can bridge DSB ends, promote short 3'-ended ssDNA ends by 3' to 5' resection from a nick, promote the loading of DSB-associated cohesin, and more. How it is implicated in SSTR and DSTR is not yet clear. Whatever steps require MRX, they do not need Sae2.

As expected, the presence of mismatches in the ssDNA template reduces the efficiency of SSTR. However, the degree of inhibition for the level of heterology we used– 4 mismatches in a 37-nt region (~11% divergence)–was only about 4-fold, quite different from the greater than the 700-fold reduction seen between dsDNA inverted repeat substrates with similar levels of heterology, but where recombination is blocked only when both Rad51 and Rad59 are deleted [84,85]. Moreover, we were surprised that the effect of these heterologies was not suppressed

by deleting either Msh2 or the helicase Sgs1, as previous studies have shown that recombination between divergent sequences–both gene conversions between dsDNA sequences and SSA–is markedly improved by deleting Sgs1 and components of mismatch repair [72,73]. For Sgs1, the 11% level of heterology is greater than that studied in SSA (3%) yet comparable to spontaneous inverted repeats (9%), but it is possible that Sgs1 cannot respond to such a level of divergence. However, we note that the ability of Sgs1 and Msh2/Msh6 to discourage recombination of heterologous sequences is much greater when the DSB end contains a nonhomologous tail versus ends that do not have such sequences (as in the cases studied here) [75]. Alternatively, Sgs1 may play a specific role in SSTR that has not yet been revealed.

Incorporation of mismatches templated by the ssODN into the genome occurs in an Msh2-dependent manner. SSTR provides an unambiguous way to distinguish between the initial strand annealing event with the one DSB end that is complementary to the ssODN and the subsequent events that lead to second end capture and the completion of DSB repair. Confronted with 4 mismatches in the ssODN on either side of the DSB, the cell efficiently incorporates these heterologies into the gene-edited product, but by two distinctly different processes. The initial annealing of the resected end with the template produces a heteroduplex DNA that should be short-lived, until DNA polymerase extends the 3' end and the newly copied strand dissociates and anneals with the second resected end. Only during this short-lived annealing step can mismatch repair transfer the heterology to the strand that will be incorporated into the final gene-edited product (Fig 7). We have previously shown that similar events occur rapidly during an HO-induced gene conversion (*MAT* switching) and depend on mismatch repair machinery [86].

Although most heterologies in the ssODN are readily incorporated, a mismatch very close to the 3' end of the ssODN is usually not incorporated, unless the 3' to 5' proofreading activity of DNA Polymerase δ is eliminated. We demonstrated a similar type of proofreading in BIR, where the 3' strand invading into a duplex DNA donor is resected [54]. By the same token, 3' to 5' resection of the DSB end will assure that a heterology close to that end (close to the *Xho*I site) could be incorporated without the need for Msh2.

Once the annealed end is extended, copying the 5' side of the ssODN, all of the heterologies on that side will be copied; but when this newly-copied strand anneals with the second DSB end, there will be an obligate heteroduplex. Hence, on this side, in the absence of Msh2, we recover sectored colonies. The failure to incorporate the 5'-most heterology may reflect dissociation of the newly copied strand before it reaches the end of the template.

These considerations lead us to the model of SSTR shown in Fig 7. After a DSB, the cell initiates end resection, forming ssDNA tails. This process may require MRX proteins to create 3'-ended tails. In the absence of this complex it is possible that neither Exo1 nor Sgs1-Rmi1--Top3-Dna2 can act soon enough to permit use of the ssDNA template before its degradation; however, MRX is required even when ssDNA templates are generated by the retron system, which presumably has the capacity to create ssDNA continually while under induction. Strand annealing depends on Rad52 and on the action of Rad59 to thwart a Rad51-mediated inhibition of Rad52's annealing activity. Polδ is also engaged in its proofreading mode to remove heterologies near either the 3' end of the DSB or the template. In SSTR, the degree of resection of the ssODN is quite limited, as only a marker 2-nt from the end is affected by this "chewing back". When strand invasion occurs during BIR, the 3' to 5' excision can extend up to about 40 bases [54]. Once the first end is annealed and extended, the newly synthesized DNA must dissociate and anneal to the second end of the DSB, thus bridging both sides of the break and creating heteroduplex DNA that is subject to mismatch repair. The final filling-in of resected ssDNA regions appears to be carried out by the translesion polymerase Polζ or other redundant polymerases. When SSTR is used to create a large deletion, long-range resection is required and the Rad1-Rad10 flap endonuclease becomes essential.

Finally, we have also determined that SSTR is a highly mutagenic event. Until now, the off target-effects of gene editing have primarily been thought of as a result of non-specific cutting of the endonuclease. Here we found that filling-in of the gaps created by long-range resection machinery following the DSB is a highly mutagenic process, dependent on DNA Polζ. It is therefore highly possible that regions adjacent to targeted DSBs have been mutated during SSTR. Gene-edited products should be screened for these potential mutations.

## Methods

### Parental strain

JKM179 (*hoΔ MATα hmlΔ*::*ADE1 hmrΔ*::*ADE1 ade1-100 leu2-3,112 lys5 trp1*::*hisG′ ura3-52 ade3*::*GAL*::*HO*) was used as the parental strain in these experiments. This strain lacks the *HML* and *HMR* donor sequences that would allow repair of a DSB at *MAT* by gene conversion, and thus all repair occurs through NHEJ or via the provided ssODN or dsODN. ORFs were deleted by replacing the target gene with a prototrophic or an antibioitic-resistance marker via the high-efficiency transformation procedure of *S. cerevisiae* with PCR fragments [87]. A list of all strains used is provided in S1 Table. Point mutations in *Rad52* (R70A mutation), *POL2 and POL3* were made via CRISPR-Cas9 with an ssODN template. gRNAs were ligated into a *BplI* digested site in a backbone that contains a constitutively active Cas9 and either an HPH or *LEU2*-marker (bRA89 and bRA90, respectively) [88]. Plasmids were verfieid by sanger sequencing (GENEWIZ) and transformed as previously described [88]. Plasmids and Cas9 donor sequences are found in S2 and S3 Tables.

### Retron plasmid construction

pZS165, a yeast centromeric plasmid marked with *ura3* for the galactose inducible expression of the retron-guide chimeric RNA with a flanking HH-HDV ribozyme [66] obtained from Addgene. gBlocks were designed that contained a gRNA and donor sequence and were cloned into the *NotI*-digested pZS165 backbone using the NEBuilder HiFi DNA Assembly Cloning kit. Integration was verified by sequencing (GENEWIZ).

### SSTR viability using HO endonuclease

Strains were grown overnight in selective media, and were then diluted into 50 mL of YEPD and grown for 3 hours. Cells were then pelleted, washed with dH$_2$O, and resuspended in 0.1M LiAc. After pelleting, 25 $\mu$L of 100 $\mu$M ssODN, 25 uL of TE, 25 $\mu$L of 2 mM salmon sperm DNA, 240 $\mu$L of 50% PEG, and 36 $\mu$L of 1M LiAc was added to the pellet and vortexed. Reactions were incubated at 30˚C for 30 minutes, followed by a 20-min incubation at 42˚C. Cells were then diluted 1000-fold and plated onto YEPD and YEP-Galactose (YEP-Gal) media. YEPD plates were grown at 30˚C for 2 days, and YEP-Gal plates were incubated at 30˚C for 3 days. Colonies were then counted to obtain average viability. SSTR vs NHEJ events were determined by pooling survivors and amplifying the *MAT* locus via PCR. Following amplification, PCR products were digested with the *Xho*I nuclease and quantified via gel electrophoresis. Data are found in S1 Table.

### SSTR viability using Cas9/Retron system

Cas9 and the Ec86 retron were integrated into *his3* on chromosome 15 into strain JKM179 that had *ade3*::*GAL*::*HO* deleted by restoring *ADE3*. After the plasmid containing the gRNA and Ec86 donor sequence was introduced, strains were resuspended in dH$_2$O and plated onto uracil drop-out media or uracil drop-out media containing galactose (URA-Gal). URA plates

were incubated at 30˚C for 3 days, and URA-Gal plates were incubated at 30˚C for 4 days. After incubation, plates were counted to obtain viability. URA-Gal plates were then replica plated to lysine drop-out media to obtain SSTR levels. Data are included in S1 Table.

### Single strand annealing analysis

SSA assays between partial *LEU2* gene repeats was previously described (Vaze et al., 2002). To determine the viability, cells were grown overnight in YEP raffinose medium (1% yeast extract, 2% peptone, 2% raffinose), and ~100 cells were plated onto YEPD and YEP-galactose plates. Colonies were counted 3–5 days after plating. The proportion of viable cells was estimated by dividing the number of colony-forming units on YEP-galactose plates by that on YEPD plates. For Southern blot analysis, DNA was isolated, digested with *Kpn*I, and separated on 0.8% agarose gels. A *LEU2* sequence was used as a probe to monitor SSA product formation, and *ACT1* probe was used as the loading control. Repair efficiency was measured as the percentage of normalized pixel intensity of the band corresponding to SSA at 6 hr in a mutant to the normalized pixel intensity of band corresponding to SSA at 6 hr in WT cells. Statistical comparison of repair efficiency between different mutants was performed using Welch's unpaired t-test.

### DNA sequence analysis

Using primers flanking the region of interest, PCR was used to amplify DNA from surviving colonies. Primers to analyze events at MAT and lys5 are listed in S2 Table. A full list of PCR primers is available upon request. PCR products were purified and Sanger-sequenced by GEN-EWIZ. The sequences were analyzed using Serial Cloner 2-6-1.

### Statistical analysis

All statistical analysis was performed using GraphPad Prism 8 software.

### Supporting information

**S1 Fig. SSTR can be determined by PCR and restriction digest with XhoI.** The percentage of SSTR after the experiment described in Fig 1 was determined by PCR across the *MAT* locus using primers DG_253 and DG_254 (S2 Table), followed by *Xho*I restriction digest. NHEJ events will result in a non-digested product of 1,674 bp, while SSTR results in two bands at 1,347 bp and 327 bp. The intensity of the bands was quantified as shown by Gel Doc Imager. Trial 1 and 2 were performed on different days with different sets of media, but with identical protocols.
(PDF)

**S2 Fig. SSA can be detected via Southern blot.** Representative southern blots showing DSB repair products by SSA in WT and r*ad52Δ* strains.
(PDF)

**S3 Fig. Deletions of translesion polymerases do not affect SSTR efficiency.** Cell viability following and HO-induced DSB with transformed ssODN with 37-nt of perfect homology and a 6-nt *Xho*I restriction site. n = 3. Error bars refer to standard error of the mean.
(PDF)

**S4 Fig. Efficiency of SSTR is different depend on ssODN template design.** Cell viability following a DSB with transformed 80-nt ssODNs that create a 6-bp insertion versus a 1-kb deletion marked by a 6-bp insertion. Viability determined by colony counts of galactose-induction media over colony counts of YEPD non-induction media. Significance determined using a

paired t-test, * p $\leq$ 0.01. *Xho*I insertion n = 19 (averaged across all assays), 1 kb deletion n = 8. Error bars refer to standard error of the mean.
(PDF)

**S5 Fig. 2-part Cas9-Retron system allows genomic manipulation.** Galactose-inducible, yeast optimized spCas9 was introduced into the trp1 locus along with a galactose-inducible, yeast optimized retron, Ec86 (RT). Upon galactose-induction apo-Cas9 and the retron are transcribed. The blue region of the ssDonor (single-stranded donor) is the donor sequence to repair the DSB break, while the red region refers to a 34-bp consensus region that the retron binds to on the mRNA transcript to initiate reverse transcription, and the yellow region represents the termination sequence. The ssDonor and the gRNA are constitutively active. Galactose-induction results in A) and irreparable DSB since no donor is encoded, or B) repair of Cas9 cleavage via the reverse transcribed retron system.
(PDF)

**S6 Fig. Cas9-Retron system show similar genetic requirements as HO at MAT.** Upon galactose-induction, a Cas9-mediated DSB is created at *MATα*, which can be repaired through a retron-genereated ssDNA template, resulting in insertion of the *Xho*I restriction site. Colonies for each mutant were plated onto URA drop-out media with dextrose (non-induction) and URA drop-out media with galactose induction media. Plates were counted and the % Viability determined by average count of induction survivors over average count on non-induction media. Significance was determined using two-tailed t-tests compared to WT, using the two-stage Benjamini, Krieger, and Yekutieli false discovery rate approach [89], * p $\leq$ 0.01, ** p $\leq$ 0.001, *** p $\leq$ 0.0001. Error bars refer to the standard error of the mean. n = 3.
(PDF)

**S1 Table. Strains used in these experiments.**
(DOCX)

**S2 Table. Oligonucleotides used in these experiments.**
(DOCX)

**S3 Table. Plasmids used in these experiments.**
(DOCX)

**S1 Dataset. All of the data concerning the efficiency of SSTR in various mutant backgrounds are presented in the accompanying dataset.**
(XLSX)

## Author Contributions

**Conceptualization:** Danielle N. Gallagher, Grzegorz Ira, James E. Haber.

**Data curation:** Danielle N. Gallagher.

**Formal analysis:** Grzegorz Ira, James E. Haber.

**Funding acquisition:** Danielle N. Gallagher, Grzegorz Ira, James E. Haber.

**Investigation:** Danielle N. Gallagher, Nhung Pham, Annie M. Tsai, Nicolas V. Janto, Jihyun Choi.

**Project administration:** Grzegorz Ira, James E. Haber.

**Supervision:** Danielle N. Gallagher, Grzegorz Ira, James E. Haber.

**Validation:** Nhung Pham, Grzegorz Ira.

**Writing – original draft:** Danielle N. Gallagher.

**Writing – review & editing:** Grzegorz Ira, James E. Haber.

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
