## [Decision Letter · Decision Letter 0]

13 Mar 2020

Dear Jim,

Thank you very much for submitting your Research Article entitled 'A Rad51-Independent P­ath­way Promotes Single-Strand Template Repair in Gene Editing' to PLOS Genetics. Your manuscript was fully evaluated at the editorial level and by three independent peer reviewers. The reviewers appreciated the attention to an important topic but identified some aspects of the manuscript that should be improved.  In addition to addressing the individual points made by each reviewer, please note that in accordance with PLOS Genetics open data policy all of the underlying numerical data for each figure in the paper needs to be provided (supplementary material is fine).

We therefore ask you to modify the manuscript according to the review recommendations before we can consider your manuscript for acceptance. Your revisions should address the specific points made by each reviewer.

[LINK]

Yours sincerely,

Gregory P. Copenhaver

Editor-in-Chief

PLOS Genetics

Gregory P. Barsh

Editor-in-Chief

PLOS Genetics

Reviewer's Responses to Questions

**Comments to the Authors:**

Reviewer #1: This manuscript uses yeast as a model to examine the genetic requirements of single-strand template repair (SSTR) using two distinct systems: (1) an HO-induced DSB repaired using an oligo introduced by transformation (Figure 1) and (2) a Cas9-induced DSB repaired by an in vivo-generated ss repair template (the CRISPEY system; Figure 3). Although the Cas9 system had a higher repair efficiency, the genetic requirements were the same as in the HO system: Rad51-independent, Rad52- and Rad59-dependent, Srs2 dependent and MRX dependent. The authors further used to the HO system to document a repair-associated increase in nearby mutagenesis of a URA3 reporter, the effects of mismatches in the during SSTR, and the effect of 500 bp of nonhomology flanking the introduced DSB. Finally they directly compared the efficiency and genetic requirements of the HO-induced break when the introduced repair template was double-stranded rather than single-stranded. The manuscript is clearly written and the data presented provide important insight into the genetic control of SSTR and how it is mechanistically different from DSTR. Finally he demonstration of SSTR-associated mutagenesis provides a cautionary note that should be that should be heeded when using SSTR for directed genome modification in higher eukaryotes. Below are comments that should be considered when revising the manuscript.

1. The SSA data do not fit well with the other analyses, so perhaps should be moved to the supplement or omitted altogether. The main thing shown is that suppression of rad59D by rad51D requires the Rad52 annealing activity. Why not just show this with the SSTR system?

2. Panels E and F in Fig. 1 are out of place – not described until after the SSA data in Fig. 2.

3. In Figure 5C, the colors of WT and the mismatch-containing ODN are reversed relative to panel A, which makes things unnecessarily confusing.

4. In Figure 5B, any idea why viability increases in the absence of proofreading activity?

5. The text in Figure 6 needs some editing: the first point under (A) is confusing and there is no “invading” strand as stated in panel (B).

6. How many repair events were sequenced for Figure 7A? If only 23 were sequenced from strain and each mismatched ODN, then the % are likely not very accurate. The error for each proportion can be determined using Vassarstats.

7. Might the asymmetry in the DSTR system reflect a transcription-generated asymmetry of the broken ends?

8. Line 459 states that the gap-filling (when creating a large deletion) depends on Pol zeta. Where was the demonstrated? There was not a rev3 mutant in Figure 4. The only thing shown to be dependent on Pol zeta was URA3-associated mutagenesis, which likely involves limited synthesis to bypass DNA damage that accumulates in ssDNA.

9. It is very striking that Sae2 has no effect in MRX-mediated resection. Any explanation?

10. In most Figures, it needs to be stated what the error bars correspond to. Also, it needs to be stated how many times were experiments repeated. How does a paired t-test differ from a normal t-test and why was it used?

11. The effect of mismatches on editing efficiency is relegated to the supplement. Why not incorporate into Figure 5 (see Figure 7B)?

12. Is it really necessary to continue showing the NHEJ in all Figures? Only SSTR or DSTR is discussed in the text.

Line 81 – delete “the fate of”

Line 117 – all survivors “in the rad52D background” resulting

Line 224 – duplex DNA?

Reviewer #2: Repair of chromosomal double-strand breaks (DSBs) using single-stranded oligonucleotide DNA templates (ssODN) is commonly used to edit genomes (single-strand template repair, SSTR). In this study, the authors use the budding yeast system to identify the mechanism and genetic requirements for SSTR.

Most of the data presented make use of an 80-nt template for repair of an HO-induced chromosomal DSB. The authors show that repair requires single-strand annealing and is independent of strand invasion factors. Interestingly, they find that the SSTR defect of the rad59 mutant is suppressed by deleting RAD51 suggesting that Rad59 is required to overcome an inhibitory effect of Rad51 on Rad52 catalyzed strand annealing. The srs2 defect, and suppression by rad51, is similar to what has been reported before for other recombination events that proceed by single-strand annealing. Importantly, the authors show that SSTR mediated by Cas9 has the same genetic requirements as repair of the HO-induced DSB.

The studies using 80-nt templates with mismatches support a model for strand annealing, extension and mismatch repair to incorporate mismatches during gene editing, consistent with a recent study in mammalian cells. Additionally, the use of the pol3-01 mutant, defective for Pol delta proofreading activity, supports the conclusion that failure to incorporate a mismatch located close of the 3’ end of the template is due to excision by Pol delta.

Overall, this is a comprehensive study documenting the genetic requirements for SSTR and will be of interest to the PLoS Genetics readership. I found the manuscript a bit disjointed because of switching between different experimental systems: Fig 1 is SSTR at HO-induced DSB; Fig 2 is SSA; Fig 3 is SSTR by Retron-Cas9; Fig 4 back to HO-induced DSB to show deletion events using a different template design and slight variation in genetic requirements; Figs 5-7 address mechanism of incorporation of mismatches; and Fig 8 presents a comparison between SSTR and DSTR. There is also a lot of repetition of genetic requirements using various assays and several errors in the Figures. I think the paper would be easier to read with a complete reorganization of how the data is presented. I have made some suggestions below for how this might be achieved.

Comments:

A previous study by Storici et al (2006) found a lower efficiency of SSTR in the rad51 rad59 double mutant than observed for the single mutants. Do the authors have an explanation for the discrepancy between the studies?

Is the requirement for MRX suppressed by eliminating Ku? This might help address whether the MRX defect is due to the resection function or another activity of the complex.

Figs 1, 3, 5, 6, 7 and 8: How many independent trials were performed for each strains? Also, the error bars need to be defined.

Fig 1E and F: these data do not really fit here and should be in a supplementary figure. Several other studies have shown mutagenesis of a nearby URA3 marker during DSB repair as a result of resection and fill-in synthesis by Pol zeta. I understand that the authors are making the point that investigators using SSTR to correct mutations in mammalian cells need to check the flanking sequence. If these are kept in it would be useful to test the distance dependence for mutagenesis.

Figs 1 and 4: Figs 1 and 4 could be combined, just showing the differences in genetic requirements when SSTR is used to create a deletion (i.e., the rad1 and exo1 sgs1 data).

Fig 2A: What is the length of the direct repeats? The SSA data appear to have been added to support the conclusion about the Rad59 role, but don't fit with the rest of the study. If Fig 2 is to be included, the authors would need to test rad52-R70A and rad51 rad52-R70A in the SSTR assay, and show the rad52 null in the SSA assay.

Fig 3C: Are the p values for comparisons of mutants to WT? What is the p value for rad59 vs rad51 rad59? Correct 80 bp donor to 80-nt.

Fig S4: It would be better to have these data included in Fig 5. The pol2-4, pol3-01, sgs1 and msh2 ssODN targeting efficiency should also be done with the homologous template.

Fig 6. The text at the bottom does not make sense as written. The left side of figure shows assimilation of the mismatches not removal. For the right panel, the statement that in the absence of Msh2 there should be no inheritance is incorrect because Fig 5 shows that mismatches can be incorporated in the msh2 mutant.

Fig 7A,B would work better incorporated into Fig. 5 with the model (Fig 6) coming afterwards.

Fig 8B, C: these are labeled SSTR instead of DSTR. There is only ~50% decrease in DSTR in the rad51 mutant; could this be due to resection of the template to make it ssDNA and therefore switching to Rad59 dependence/Rad51 independence? I think it would be more appropriate to have the SSTR/DSTR comparison included in Fig 1 or as Fig 2.

Fig 8E, F: These data could be removed, they are inconclusive and do not add to the conclusions of the study.

Line 381: Could the authors explain what they mean by “distinct from that involving gene conversion between long regions of homology”?

Reviewer #3: Gallagher and colleagues perform a highly timely study on the genetic requirements of SSTR in yeast. They found that SSTR relies completely on Rad52 and also on Rad59, proteins involved in annealing, but is independent of other Rad52 epistasis group members involved in double strand break repair - Rad51/54/55 – in fact SSTR is suppressed by Rad51 overexpression. Mre11 and Rad50 are also quite important for SSTR, but not MRX resection partner Sae2. Srs2, which strips Rad51 from single-stranded DNA, is also important; thus, mutating rad51 restores SSTR to an srs2 mutant. In fact, SSTR dramatically increases in rad51rad59 as well, suggesting to the authors that Rad59 partially alleviates the inhibition of Rad51 on Rad52 annealing. The genetic requirements are somewhat different between simple insertion of a restriction site and deletion of a larger chromosomal segment (eg the latter has a dependence on resection factors and Rad1). Mutagenesis of a nearby gene is hugely increased (600-fold). Mismatch incorporation and studies in msh2 provide key mechanistic insight.

As the authors point out, in terms of mechanistic studies of DSB repair, “budding yeast still serves as an important platform”. This manuscript shows why in that numerous factors were interrogated and experimental designed varied. Although some of the results are not surprising, given results in a few previous studies (e.g.,the comprehensive nature of this study provides a satisfying understanding of DSB repair using an ssODN. Improving this approach is a major goal of gene editors, and so this manuscript will be a welcome addition to the field to set a foundation.

Specific comments:

1. In Fig 1, the statistical (in)significance is confusing. For sgs1 and rad59, mutation of either seems to reduce SSTR substantially, yet “in subsequent assays Rad59 proved to be required”. On the other hand, Sgs1 is concluded not to be important, although rad59 and sgs1 mutants both seem to have -fold reductions in SSTR. Also, mre11 does not appear to be significant, despite the large effect and reasonable small error bars (SD or SEM). How many experiments were performed?

2. Although the SSA assay is useful (Fig 2), it is difficult to extrapolate to SSTR. For example, SSTR is increased 4-fold in the rad51rad59 double mutant, but SSA is only rescued somewhat. Is there a reason why SSTR was not examined in the rad52-R270A mutant, to try to address the genetic relationship? The authors should consider moving this figure to the supplement, since it is something of a distraction.

3. The retron results are impressive. However, the authors conclude that the SSTR genetic requirements are identical using Cas9 and HO (line 189). While broadly similar (e.g., Rad52 dependence; Rad51 independence), there are some clear distinctions (Fig 3C). For example the dependence for on Srs2 is greatly reduced and the effects of rad51 on rad59 and rdh54 are greatly muted. There are other key differences in the analysis with Cas9 besides the nuclease, eg, use of the retron system. Therefore, the conclusion needs to be tempered. The authors should directly compare Cas9 and HO using the same ssODN approach at the same locus to get some understanding of what the differences are. (Is it not possible to target Cas9 – or a Cas9 with a modified PAM - to the HO site?)

4. Fig 1E. What happens to mutagenesis frequency if the URA3 gene is placed 3’ to the ssODN site? Or if a dsDNA template is used?

5. For greater clarity, a suggestion would be to reorganize Figures 5 and 6 such that each figure contains one substrate (5’MM or 3’MM), the observed products, and the models for wildtype and for msh2. For example, with 5’MM, it would seem like MSH2 works on both strands, given the predominance of heteroduplex products in the mutant and the presence of a substantial number of products with both 3 (10 products) and no incorporated polymorphisms (7 products), but that is not apparent in the model.

6. The authors need to discuss the new results from yeast with previously published results. Although the references are there, a coherent discussion is lacking.

Additional comments:

Abstract: The authors should consider a punchier abstract to more succinctly get across key points.

Line 104: It would be helpful to indicate here, rather than below, the 10-fold improved survival by adding the ssODN: 2.8% (line 108) vs 0.2% (line 97), although NHEJ rang from 10-25% (line 104-105)

Fig 1B. Error bars should be added for the NHEJ events in each bar.

Line 121: Specify which Rad51-independent pathways.

Line 124: The distinction for Sae2 is interesting. What is the Sae2 requirement for other Rad51-independent pathways?

Line 130: It’s clear that rad51rdh54 are not epistatic (Fig. 1D); however, the epistasis analysis was not done together with srs2 (Fig. 1C) the focus of this section.

Line 131 rad51rdh54 (Fig 1D): 4-fold not 7-fold? If 7-fold, please clarify.

Line 142: It would appear that deletion of Rad59 has a much smaller effect than rad52 R270A (Fig. 2) not the “similar” effect that is stated. Is the difference not significant, despite appearances?

Line 154: “Deleting Sgs1 had no effect” same comment as above. Also, this is Fig. 1C not 1D.

Fig 5: the color scheme is confusing. In A the ssODN is black and the chromosome red, but in C the ssODN sequences incorporated are red.

Line 308: Only the mutation 2-nt from the 3’ end of the donor is said to be incorporated at a higher frequency in the pol3-01 mutant. However, it appears that that mutations in the 5’MM are incorporated at a somewhat higher rate. Is this not significant?

Line 312: it’s not clear why the authors suggest that the ssODN could be directly incorporated in some cases. Why not just propose a model as in 6B where the last mutation is maintained in a pol3-01 background now simply be copied like the rest of the 3’ mutations?

Line 326: Please explain what inverted thymine/dideoxythymines are.

Line 346: Is another acronym necessary since DSTR seems to have the genetic dependencies of classical HR?

Fig 8C,D. are the panels mislabeled – SSTR?

Line 357: it’s confusing to switch to “9 bp upstream” or “9 bp downstream” when the mutations are specifically numbered.

Line 363: Given the ambiguity, consideration should be given to moving these figure panels (8D,E) to the supplement.

Line 426: The language is not always precise and this is one example. “The initial annealing of the resected end with the template…” There are two resected ends and moreover the authors shift between ssODN, template, and donor”. Why not change around the sentence to “The initial annealing of the ssODN with the complementary resected end…”

Fig. 6: What prevents extension of the donor? Or is this considered irrelevant since it would be dissociated?

Line 438: inconsistent “…all of the heterologies will be copied” but line 441 “The failure to incorporate..” Delete “all of”

Line 454: Fig 8 not Fig 7?

**Have all data underlying the figures and results presented in the manuscript been provided?**

Reviewer #1: Yes

Reviewer #2: Yes

Reviewer #3: No: A data file should be made available for the primary data for each of the figure panels (Unless I overlooked it)

PLOS authors have the option to publish the peer review history of their article (what does this mean?). If published, this will include your full peer review and any attached files.

Reviewer #1: No

Reviewer #2: No

Reviewer #3: No

---

## [Editor Report · Decision Letter 1]

3 Aug 2020

Dear Jim,

We are pleased to inform you that your manuscript entitled "A Rad51-Independent P­ath­way Promotes Single-Strand Template Repair in Gene Editing" has been editorially accepted for publication in PLOS Genetics. Congratulations!

Yours sincerely,

Gregory P. Copenhaver

Editor-in-Chief

PLOS Genetics

Gregory Barsh

Editor-in-Chief

PLOS Genetics

Comments from the reviewers (if applicable):

**Data Deposition**

http://datadryad.org/submit?journalID=pgenetics&manu=PGENETICS-D-20-00253R1

**Press Queries**

---

## [Editor Report · Acceptance letter]

29 Sep 2020

PGENETICS-D-20-00253R1 

A Rad51-Independent P­ath­way Promotes Single-Strand Template Repair in Gene Editing 

Dear Dr Haber, 

We are pleased to inform you that your manuscript entitled "A Rad51-Independent P­ath­way Promotes Single-Strand Template Repair in Gene Editing" has been formally accepted for publication in PLOS Genetics! Your manuscript is now with our production department and you will be notified of the publication date in due course.

With kind regards,

Matt Lyles

PLOS Genetics

On behalf of:
